# Imbalance in *Unc80* RNA Editing Disrupts Dynamic Neuronal Activity and Olfactory Perception

**DOI:** 10.3390/ijms25115985

**Published:** 2024-05-30

**Authors:** Hui-Wen Chen, Chung-Pei Ma, En Chin, Yi-Tung Chen, Teh-Cheng Wang, Yu-Ping Kuo, Chia-Hao Su, Po-Jung Huang, Bertrand Chin-Ming Tan

**Affiliations:** 1Department of Biomedical Sciences, College of Medicine, Chang Gung University, Taoyuan 333, Taiwan; leggierocresc@gmail.com (H.-W.C.); ksoksx@gmail.com (C.-P.M.); a50924angel@gmail.com (E.C.); love5p35p3@gmail.com (Y.-T.C.); pjhuang@gap.cgu.edu.tw (P.-J.H.); 2Graduate Institute of Biomedical Sciences, College of Medicine, Chang Gung University, Taoyuan 333, Taiwan; tomking991@gmail.com; 3Molecular Medicine Research Center, Chang Gung University, Taoyuan 333, Taiwan; ellen0918@gmail.com; 4Center for General Education, Chang Gung University, Taoyuan 333, Taiwan; chiralsu@gmail.com; 5Department of Radiation Oncology, Kaohsiung Chang Gung Memorial Hospital, Kaohsiung 833, Taiwan; 6Genomic Medicine Core Laboratory, Lin-Kou Medical Center, Chang Gung Memorial Hospital, Taoyuan 333, Taiwan; 7Division of Colon and Rectal Surgery, Lin-Kou Medical Center, Chang Gung Memorial Hospital, Taoyuan 333, Taiwan; 8Department of Neurosurgery, Lin-Kou Medical Center, Chang Gung Memorial Hospital, Taoyuan 333, Taiwan

**Keywords:** RNA editing, *Unc80*, olfactory perception, neuronal activity

## Abstract

A-to-I RNA editing, catalyzed by the ADAR protein family, significantly contributes to the diversity and adaptability of mammalian RNA signatures, aligning with developmental and physiological needs. Yet, the functions of many editing sites are still to be defined. The *Unc80* gene stands out in this context due to its brain-specific expression and the evolutionary conservation of its codon-altering editing event. The precise biological functions of *Unc80* and its editing, however, are still largely undefined. In this study, we first demonstrated that *Unc80* editing occurs in an ADAR2-dependent manner and is exclusive to the brain. By employing the CRISPR/Cas9 system to generate *Unc80* knock-in mouse models that replicate the natural editing variations, our findings revealed that mice with the “gain-of-editing” variant (*Unc80*^G/G^) exhibit heightened basal neuronal activity in critical olfactory regions, compared to the “loss-of-editing” (*Unc80*^S/S^) counterparts. Moreover, an increase in glutamate levels was observed in the olfactory bulbs of *Unc80*^G/G^ mice, indicating altered neurotransmitter dynamics. Behavioral analysis of odor detection revealed distinctive responses to novel odors—both *Unc80* deficient (*Unc80*^+/^**^−^**) and *Unc80*^S/S^ mice demonstrated prolonged exploration times and heightened dishabituation responses. Further elucidating the olfactory connection of *Unc80* editing, transcriptomic analysis of the olfactory bulb identified significant alterations in gene expression that corroborate the behavioral and physiological findings. Collectively, our research advances the understanding of *Unc80*’s neurophysiological functions and the impact of its editing on the olfactory sensory system, shedding light on the intricate molecular underpinnings of olfactory perception and neuronal activity.

## 1. Introduction

RNA editing is a co-transcriptional mechanism that modifies specific nucleotides within an RNA molecule, altering its sequence after being transcribed from DNA. This process can occur at various sites within the RNA, leading to multiple functional outcomes, such as changes in the amino acid sequence of the encoded protein and modifications to transcript stability [1]. Notably, A-to-I editing by the ADAR2 enzyme often results in amino acid substitutions that diversify protein isoforms, playing a critical role in neurotransmission [2,3]. This includes the editing of ligand-gated and voltage-gated ion channels, as well as G protein-coupled receptors within the nervous system. A prime example of such editing involves the mRNA for GluA2, an AMPA receptor subunit essential for fast excitatory synaptic transmission in the brain. The precise editing of GluA2 is crucial for neuronal survival, preventing excessive calcium influx that can lead to the death of neurons [4] and motor neurons [5,6]. Engineering the essential edits into the *Gria2* (the GluA2-encoding gene) of *Adar2*-knockout (KO) mice has been shown to prevent mortality, indicating the critical nature of this editing event [7]. However, even with developmental rescue, these mice exhibit subtle but varied phenotypes [8], the basis of which remains to be fully elucidated, including the related substrates and mechanisms.

In the field of behavioral neuroscience, RNA editing is recognized for its involvement in various critical processes, such as synapse formation [9], ion channel regulation [10], and the modulation of neurotransmitter release [11,12]. This editing plays a vital role in the development and functioning of the central nervous system, particularly in modulating the activity of specific neurotransmitter receptors [2,3]. Studies have also highlighted the importance of RNA editing in the operation of certain brain regions and neural circuits that govern behavior. As a result, disruptions in RNA editing processes are associated with the onset of numerous neurological and psychiatric conditions, including Alzheimer’s disease [13], Parkinson’s disease [14], and schizophrenia [15,16]. Such irregularities in RNA editing are closely linked to the development of these disorders, underscoring their significant physiological roles in the central nervous system [17]. Intriguingly, RNA editing of the 5-HT2C receptor influences its cell surface expression by affecting the efficiency of intracellular trafficking, thereby regulating the density of 5-HT2C receptor binding sites in the brain [18]. Consequently, this editing event is pivotal for various physiological and behavioral functions, including circadian rhythms, emotional regulation, and appetite control [11,12,13].

Interestingly, *Unc80* has been recognized by several deep-sequencing studies, including ours, as one of many RNA-recoding targets [19,20,21]. A notable A-to-I editing event within *Unc80* is believed to recode the Ser2732 to a Gly residue in Unc80’s primary sequence, suggesting that this specific RNA alteration could influence the structure and functionality of this novel protein. Encoded by a 45-exon gene, Unc80 is a substantial protein comprising 3300 amino acids and is predominantly expressed in neurons. This gene is highly conserved, with functional homologs identified in *C. elegans* and fruit flies, and reportedly associated with the maintenance of neuronal networks [22,23,24]. Functionally, UNC80, in concert with UNC79, serves as a scaffold for Src Family Kinases (SFKs) and interacts with the NALCN channel to form a sodium-leak channel complex linked to G protein-coupled receptors [25,26]. As a critical component of the UNC79-UNC80-NALCN complex, UNC80 facilitates NALCN’s function in basal sodium leak conductance in neurons, thereby modulating neuronal excitability through extracellular calcium ions. Structural studies, such as cryo-EM analysis of the NALCN complex, have revealed that UNC79 and UNC80 form a substantial heterodimer that is crucial for the correct cellular localization of NALCN [27].

Developmentally, the absence of *Unc80* results in severe apnea at birth, mirroring some defects seen in *Nalcn* knockout models [28]. In humans, mutations in *UNC80* are linked to profound neurological and cognitive issues, including hypotonia and severe intellectual disabilities [29,30,31], echoing the critical developmental role of *Unc80* observed in mouse models. Collectively, these findings emphasize *Unc80*’s essential regulatory function in neuronal signaling and networks, although the unique phenotypes observed in *Unc80* knockout mice highlight the need for further investigation into its diverse roles [28]. Intriguingly, mutations in *Unc80* have been shown to impact various biological processes. For instance, in honeybees, differences in *Unc80* expression, observed through NGS-Seq of brains from odor-learning tasks, suggest a correlation between *Unc80* and olfactory learning [32]. In *C. elegans*, mutations in *Unc80* led to impaired avoidance behavior to Methyl salicylate, which could be significantly mitigated by neuron-specific transgenic rescue, further underscoring *Unc80*’s role in neural function [33].

Our previous work has identified a significant codon-altering RNA editing event in the *Unc80* gene through deep sequencing of the RNA editome (annotation of RNA editing events) in mammals [34]. The physiological and developmental roles of *Unc80* and its editing, which are both brain-specific and evolutionarily conserved, are yet to be fully understood. To explore these aspects, we have engineered *Unc80* knockout mice and mice with loss- or gain-of-editing in *Unc80* using the CRISPR/Cas-9 system. Our study provides insights into the function and regulation of *Unc80* and its editing, particularly in relation to olfactory function and the associated molecular mechanisms.

## 2. Results

### 2.1. Characterization of the Expression and RNA Editing Event of Unc80

Among potential candidate editing events uncovered in our previous high-throughput sequencing approach [34], *Unc80* was chosen for its notable features: (1) its editing is evolutionarily conserved among mammals; (2) both *Unc80* and its RNA editing are predominantly found in the brain (Figure 1A,B), indicating their significant roles in neuronal processes; (3) Sanger sequencing of tissues from specific brain regions confirmed that this editing event relies on the ADAR2 enzyme, as evidenced quantitatively by the absence of the A-to-G conversion in *Adar2* knockout mice. In addition, the *Unc80* editing was particularly abundant in the cerebellum and olfactory bulb in wild-type mice, directing our research towards a detailed examination of *Unc80* and its editing within these brain areas (Figure 1B). To explore whether the editing event occurring in the coding sequence of Unc80 caused any changes to the protein, we applied a computational method, AlphaFold2 [35], to predict the possible conformational change due to the Ser to Gly substitution at residue 2732. We systematically modeled the 3-dimensional protein structures of the full-length Unc80 (3267 amino acids) (Appendix A) and used PyMOL [36] for subsequent analysis to visualize and analyze the detailed protein structures between Ser2732 and Gly2732 as well as the nearby amino acid residues (Appendix A). Interestingly, our results indicated a secondary structure change in this specific region: compared to the wild-type protein, the secondary structure, especially around this editing region, has changed from a loop to approximately two turns of a helix, with the editing residue Gly2732 as the center (Figure 1C). This change may lead to alterations in the function and properties of the protein or how the protein interacts with other molecules within the cell. 

Based on the findings from honeybees and *C. elegans* [32,33], and in line with our data, we further explored the physiological importance of Unc80 by examining its expression patterns, especially in the olfactory bulb, where the editing rate was higher. The localization of Unc80 showed significant enrichment in the mitral cell layer (MCL) and rostral migratory stream (RMS), critical components of a functional olfactory bulb. Some of these signals overlapped spatially with the expression of NeuN, a neuronal cell marker, thus establishing the link of Unc80 to odor perception (Figure 1D). Moreover, we also set out to examine and confirm *Unc80* editing in the primary recipients of olfactory bulb (OB) outputs: the anterior olfactory nucleus (AON) and the anterior piriform cortex (APC), both of which are vital for odor perception and forming the anterior olfactory cortex (AOC) [37,38,39]. As the sequence chromatograms showed in Appendix A, the same editing event was verified quantitatively in AON and APC.

### 2.2. Engineering and Phenotyping of the Unc80 Knockout and Knock-In Mouse Models

To elucidate the biological implications of the *Unc80* gene product and the associated gene recoding event, we set out to create mouse models with (1) a deletion of the gene and (2) an A-to-G point mutation at the editing site (see schematics in Figure 2A for experimental workflow). Utilizing the CRISPR/Cas9 system and the detailed experimental scheme we performed (Figure 2A), we successfully established engineered mice with a genomic deletion and site-specific A-to-G mutations at the editing locus of the *Unc80* gene (corresponding to AGC of the Ser2732 codon), resulting in “gain-of-editing” (AGC→GGC; *Unc80^G/G^*) and “loss-of-editing” (codon-neutral mutation, AGC→TCC; *Unc80^S/S^*) knock-in models for subsequent breeding (Figure 2B,C). Initial evaluations of these mice have revealed several notable phenotypes. When heterozygous mice with the deletion allele were crossed, the resulting homozygous deletion offspring were born at a significantly lower frequency than Mendelian genetics would predict (Figure 2D,E), indicating partial embryonic lethality and growth retardation, which suggests that *Unc80* is crucial for normal development. Analysis of *Unc80*-deficient mouse brains through immunoblotting and qRT-PCR showed no significant change in the levels of transcripts encoding Nalcn, a protein that interacts with Unc80 (Figure 2F,G). In addition, to elucidate whether the genetic manipulation of the coding sequence may affect the proper localization of the protein, we first performed olfactory bulb immunohistochemical (IHC) staining on *Unc80^S/S^* and *Unc80^G/G^* mice (Appendix A). According to the images, the distribution of Unc80 protein expression in the olfactory bulb showed no difference between *Unc80^S/S^* and *Unc80^G/G^* mice. Secondly, we transfected specific GFP-tagged wild-type Unc80 or Unc80^S2732G^ encoding plasmids into a mouse neuroblastoma cell line (Neuro 2a cells) and performed immunofluorescence microscopy analyses to verify the localization of the fusion protein. By following the GFP signals, our results showed that both wild-type Unc80 and Unc80^S2732G^ exhibited the same localization inside the cell, predominantly in the cytosol (Appendix A).

### 2.3. Neuronal Activity in the Unc80 Animal Models Revealed by MRI

The severe clinical manifestations observed in individuals with *Unc80* mutations suggest that *Unc80* is a critical regulator in neuronal signaling and networks [40]. To investigate the neuronal activity in the olfactory system, we applied in vivo MR spectroscopy in *Unc80*-targeted animal models to map neural activation patterns and assess neurotransmitter levels, including glutamate and dopamine, in the brain. Additionally, we utilized manganese-enhanced MRI (MEMRI), a method for imaging neuronal activity through systemic administration of manganese ions, which act as a T1-shortening contrast agent. Specifically, during functional stimulation, the manganese ions (Mn^2+^) enter active neurons via voltage-gated calcium channels [41,42]. Following sustained stimulation, Mn^2+^ accumulates in activated brain regions, resulting in enhanced signals on T1-weighted imaging. Toward this end, we utilized MEMRI to monitor neuronal activity within the olfactory system. This technique involves the systemic administration of Mn^2+^ ions as a T1-shortening contrast agent to delineate the activity within the olfactory tract, including the olfactory bulb (OB), anterior olfactory nucleus (AON), and anterior piriform cortex (APC). These imaging modalities aimed to uncover any signs of brain pathology in *Unc80* mouse models. Our findings indicated an increased MEMRI signal intensity in the selected nuclei of the gain-of-editing *Unc80^G/G^* mice, suggesting enhanced neuronal activity along the olfactory tract (Figure 3A,B). Furthermore, we utilized chemical exchange saturation transfer (CEST)-MRI to quantitatively map the distribution and influx of neurotransmitters, such as glutamate and dopamine, in specific brain areas. This functional imaging revealed significant differences in glutamate levels between *Unc80^S/S^* and *Unc80^G/G^* mice, particularly in the olfactory bulbs and hippocampus (Figure 3C,D). This altered resting neurotransmission reinforces the central role of *Unc80* RNA recoding in olfactory perception.

### 2.4. Implication of Unc80 Editing Event in Mediating Olfactory Response

To understand the impact of differential *Unc80* editing on neuronal activity in the olfactory bulbs and its potential effects on olfactory responses, we conducted an odor-evoked sniffing test. This test assessed the mice’s ability to detect and differentiate between various odors. We measured the degree of odor habituation (sensitization) and dishabituation (desensitization) across successive 7 rounds of odor exposure, along with the latency to habituation. Additionally, an eighth trial introduced a new odor to evaluate re-sensitization and the potential reversal of habituation. Intriguingly, *Unc80* heterozygous mice (*Unc80^+/−^*) demonstrated a more pronounced decrease in odor sensitivity over the initial seven trials compared to wild-type mice, which showed a more gradual habituation process (Figure 4A). Furthermore, when presented with a new odor, the response of *Unc80* heterozygous mice was significantly heightened relative to that of wild-type mice, indicating increased time spent investigating the novel odor source. These findings thus suggest that a deficiency in *Unc80* may lead to an overly sensitized olfactory response characterized by increased dishabituation. Notably, *Unc80^S/S^* mice, which lack the editing modification, exhibited a habituation/dishabituation pattern similar to that of the *Unc80* heterozygous mice, hinting that the absence of editing might similarly affect olfactory perception (Figure 4B).

For the odor-induced neuronal activity assay, we used the strong scent of banana oil (isoamyl acetate) to stimulate nerve signals in the olfactory bulbs, as indicated by c-Fos expression [43]. Mice with or without *Unc80* editing were tested for differences in odor-sensing nerve stimulation. After a 30 min exposure to banana oil, mice were sacrificed either immediately or at 60 min post-exposure. The olfactory bulbs were then collected for gene expression analysis via qRT-PCR and immunoblotting. Our results (Figure 4C,D) showed that *Unc80^S/S^* mice exhibited a significantly greater increase in c-Fos expression immediately following odor exposure compared to *Unc80^G/G^* mice, suggesting differing sensitivities to external odor stimuli. Moreover, immunoblotting of olfactory bulbs for c-Fos protein expression revealed a substantial increase in *Unc80^S/S^* mice in response to odor stimuli compared to *Unc80^G/G^* mice, which already exhibited a high baseline level of c-Fos signal (Figure 4D). This enhanced induction of c-Fos in response to odors may be influenced by the editing state of *Unc80*.

### 2.5. Transcriptome Profiling of Unc80 Knock-In Mice Revealed Altered Neuronal State in Olfactory Bulb

After phenotyping the *Unc80* editing knock-in mouse model, we next aimed to substantiate the biological significance by exploring possible molecular perturbations resulting from the Unc80 recoding event. In this regard, we conducted RNA-seq-based profiling to examine transcriptome-wide alterations in the olfactory bulbs of *Unc80^S/S^* and *Unc80^G/G^* mice, identifying molecular changes triggered by the editing event. The overall distribution of transcriptome profiles, illustrated by a PCA plot (Figure 5A) and a volcano plot (Figure 5B), revealed genotype-specific, significant changes in gene expression. A total of 70 genes exhibited substantial expression differences between the transgenic *Unc80^S/S^* and *Unc80^G/G^* mice, with 27 genes upregulated and 43 genes downregulated. The distinct clusters of differentially expressed genes showed at least a 1.5-fold change for all significant DEGs (Figure 5C). Subsequently, we used Ingenuity Pathway Analysis (IPA, version Spring Release Q1 2024) to explore the functional enrichment pathway contributed by DEGs. The bubble chart indicated that several neuron-related categories were highly enriched, including cellular growth, proliferation and development, neurotransmitters and other nervous system signaling, transcriptional regulation, organismal growth and development, and intracellular and second messenger signaling (Figure 5D and Appendix A). In addition, according to previous studies, many of these are closely related to brain development, the olfactory system, neuronal activity, and neuro-related diseases. This includes three functional categories: G protein-coupled receptor (GPCR), ion channels, and transcription factors. Specifically, the expression levels of genes associated with GPCRs, such as Nkain3 and Gpsm3, showed notable differences between the *Unc80^S/S^* and *Unc80^G/G^* mice. Transcription factors known to regulate neuronal activity, such as Nr4a2 and FosB, also displayed significant expression changes between these mouse models [44,45]. These results suggest that the phenotypic outcomes of Unc80 recoding are associated with distinct signaling alterations, including enrichment in ERK-associated, cAMP-mediated, and neuronal-related signaling pathways, along with the expression of unique neuronal gene signatures (Figure 5E). These molecular insights lay a crucial foundation for our mechanistic understanding of this editing event.

## 3. Discussion

In the context of odor stimulation, honeybees with *Unc80* deletions exhibit altered odor-evoked responses, along with changes in their transcriptome. Similarly, in C. elegans, the absence of *Unc80* impairs the avoidance response to methyl salicylate, a phenotype that is reversible with the restoration of *Unc80*. These observations from olfactory behavior analyses suggest a critical link between *Unc80* and the olfactory system. Furthermore, the involvement of Unc80 in a voltage-independent ‘leak’ ion-channel complex, which is extensively regulated by neurotransmitters and GPCR activation, underscores its distinctive role in odor discrimination. Nevertheless, the neurobiological outcomes and mechanisms by which *Unc80* influences the sense of smell remain largely unexplored. In our study, building upon our preliminary findings, we initially mapped the distinctive *Unc80* editing patterns explicitly expressed in the brain, particularly in the olfactory bulb area (Figure 1B). Subsequently, to investigate the effects of mRNA recoding of *Unc80* in an in vivo model, we generated transgenic knock-in mice either lacking (*Unc80*^S/S^) or expressing the recoding event of Unc80 (*Unc80*^G/G^) using the CRISPR/Cas9-based method (Figure 2A). By employing these transgenic mice in anatomical and functional MRI analyses, olfactory behavior tests, and high-throughput transcriptomic approaches, we have highlighted its significance in regulating olfactory function. This work has begun to unravel a comprehensive network of genes involved in brain development, olfactory function, and neurodegenerative diseases.

Our results have demonstrated a regulatory role for *Unc80* editing in odor detection. However, the notion that the NACLN complex is also involved in this physiological context is not yet clear. Previous studies have shown that Unc80 and Unc79 in the brain are crucial for the formation of the NALCN complex, which is integral to calcium channel function [46,47]. This suggests that Unc80 may function in a manner dependent on Unc79. Remarkably, homozygous *Unc80* knockout pups display severe apnea shortly after birth, a phenotype also observed in NALCN knockout models, suggesting similar functional roles [28,48]. However, clinical data from the MalaCards human disease database indicates that certain pathologies associated with *Unc80* deficiency are distinct from those arising from mutations in *Nalcn* or *Unc79*. For instance, issues such as mitochondrial dysfunction and neurodegenerative diseases, like ALS, are uniquely associated with *UNC80* loss [49,50]. This implies that UNC80 may also have functions independent of the NALCN complex, which have yet to be explored. Moreover, the severe phenotypes observed both in mouse models with targeted *Unc80* disruption and in humans with *UNC80* mutations underscore its critical role in neurodevelopment [28]. The distinct genetic abnormalities associated with *UNC80* hint at a unique molecular function that could be independent of its established interaction with the NALCN complex, suggesting additional yet-to-be-identified roles for UNC80 in cellular functions.

One of the critical findings in the study is to explore the comprehensive gene expression profile of olfactory bulbs in mice harboring the *Unc80* editing site, compared to those without it. The wide spectrum of gene regulation mediated by *Unc80* editing is consistent with the notion that the recoding of Unc80 impacts the olfactory system by modulating a network of genes closely associated with olfactory functions. Significantly altered genes were categorized into three groups: channels and receptors, GPCRs, and transcription factors. The channel and receptor group includes genes related to neuronal signaling, such as Pkd2L1, a transient receptor potential cation channel implicated in recovery from spinal cord injury [51,52]. Moreover, the sensation of smell in mice is facilitated by a multitude of chemosensory receptors. Approximately 1,400 olfactory receptors, which form one of the largest families of GPCRs, are expressed on olfactory sensory neurons and are essential for olfactory discrimination [53,54,55,56]. Upon odorant binding, these receptors trigger GPCR signaling, leading to a cascade of signal transduction. Thus, the targets of *Unc80* editing we identified may provide a foundation for elucidating the mechanisms underlying olfactory perception. Moreover, transcription factors affected by *Unc80* editing play a vital role in determining the sensory identity of olfactory receptor neurons. Incidentally, specific transcription factors, such as Nr4a2 and FosB, have been implicated in the processing of odor information, with alterations in their expression correlating with early olfactory dysfunction in Alzheimer’s disease mouse models [45]. In conclusion, through detailed transcriptomic analysis, *Unc80* editing emerges as a key regulator of transcriptomic changes within the olfactory bulb, establishing a new framework for understanding the functional implications of Unc80 recoding.

The olfactory perception pathway initiates when an odorant binds to its receptor, triggering a GPCR and subsequently causing a rise in intracellular cAMP concentration. This increase in cAMP generates an action potential through ion influx, leading to neuronal transduction [57,58,59,60]. Our high-throughput data suggest a signaling output tied to Unc80-mediated neuronal regulation, potentially involved in neuronal differentiation or electrophysiological alterations, signifying cellular changes at the molecular level. One particular gene, Zebd6, shows a significant change in expression in our dataset and is known to play a role in metabolism and signal transduction pathways [61,62,63]. However, its specific functions in the olfactory system and the associated signaling cascades remain undefined. Given the intricacies of upstream regulation, it is plausible that the signaling cascade dynamically responds to changes in the expression or editing of intracellular Unc80. Our research delineates novel regulatory functions of *Unc80* and its RNA editing in olfactory perception, presenting the first evidence of a previously unrecognized association between *Unc80* editing and olfactory function. This link opens up new avenues for understanding sensory processes’ complexity and underlying molecular mechanisms.

## 4. Materials and Methods

### 4.1. CRISPR-Mediated Genome Editing for Generation of Point Mutant Mice

The CRISPR/Cas9 system was used to engineer *Unc80* editing site mutations in vivo in wild-type mice. Briefly, the linearized T7-Cas9 and T7-sgRNA PCR products were gel purified and used as the templates for in vitro transcription (IVT) using mMESSAGE mMACHINE T7 ULTRA kit (Thermo Fisher, Waltham, MA, USA) and MEGAshortscript T7 kit (Thermo Fisher), respectively. The resulting Cas9 mRNA and the sgRNAs were purified using a MEGAclear kit (Thermo Fisher) and eluted in RNase-free water. B6D2F1 female mice and ICR mouse strains were used as embryo donors and foster mothers, respectively. Pre-determined concentrations of Cas9 mRNA, sgRNA, and oligonucleotides were mixed and co-injected into the cytoplasm or pronucleus of fertilized eggs with well-recognized pronuclei in the M2 medium (Sigma-Aldrich, Saint Louis, MO, USA). Injected zygotes were cultured until the blastocyst stage by 3.5 days and then transferred to the uterus of pseudopregnant ICR females. The single-stranded DNA oligonucleotide donor carrying “pre-edited” alleles or silent mutations disrupting the editing substrate was synthesized for co-injection.

### 4.2. Animals

Mice had ad libitum access to a standard diet and were housed in a pathogen-free facility under a 12 h light/dark cycle. All breeding and experimental procedures were conducted following IACUC guidelines at Chang Gung University, and the approval was CGU111-164. For experimental purposes, male mice aged 2–5 months were utilized.

### 4.3. RNA Extraction, RT-PCR, and qPCR 

RNA from specified tissues was extracted using TRIzol reagent (Invitrogen, Carlsbad, CA, USA) and reverse-transcribed into cDNA using MML-V reverse transcriptase (Invitrogen) with random hexamers according to the manufacturer’s instructions. The *Unc80* editing ratio was monitored by Sanger sequencing of PCR-amplified products using primers for targeted regions. Gene expression was quantified by real-time PCR (Bio-Rad CFX Connect Real-time system) with specific primers and analyzed with CFX Manager Software version 3.1 (Bio-Rad, Hercules, CA, USA). Amplification was performed with SYBR^®^ Green Master Mix (Bio-Rad), starting with a 3-min heat at 95 °C, followed by 40 cycles (95 °C for 10 s, 60 °C for 30 s), with a final dissociation curve stage. Relative gene expression was normalized to internal control genes, with controls as reference. All results were obtained from at least three independent experiments, presented as mean ± SEM, and statistically evaluated using Student’s *t*-test.

### 4.4. Immunofluorescent Staining of Brain Tissue 

Mice were transcardially perfused with cold phosphate-buffered solution (PBS) followed by 4% paraformaldehyde after deep anesthesia. The whole brains were fixed in 4% paraformaldehyde for 48 hours and then stored in 30% sucrose solution for 48 hours. Sagittal frozen sections (30 μm) were cut with a cryostat (Leica Microsystems, Wetzlar, Germany, SM 2010R), permeabilized, and blocked with 10% BSA in 0.5% Triton X-100 for 2 h, then incubated overnight at 4 °C with primary antibodies against Unc80 (BS-12121R, BIOSS, Woburn, MA, USA) and NeuN (MAB377, Merck Millipore, Darmstadt, Germany). Images of the slices were acquired with a fluorescence microscope (BioTek Lionheart FX Automated Microscope, Agilent, Santa Clara, CA, USA). 

### 4.5. Manganese-Enhanced Magnetic Resonance Imaging (MEMRI) 

The MRI technique involves placing anesthetized mice in a prone position within an acrylic holder inside a cryo probe coil for MRI scans. The spin echo-planar imaging DTI sequence, covering ten 200 μm coronal slices, matches the spatial dimensions of T2-weighted reference images. Acquisition parameters such as field of view, matrix dimensions, spatial resolution, and echo time are tailored to the experiment. Post-processing includes fiber tractography and DTI index analysis using DSI studio software (http://dsi-studio.labsolver.Org). Specifically, fiber tracts between both hippocampi are evaluated, and regions-of-interest (ROIs) analyses quantify DTI indices like fractional anisotropy (FA) and fiber tract numbers. For MEMRI, manganese (Mn^2+^), a natural cellular constituent, serves as a T1-shortening MR contrast agent due to its paramagnetic properties and cellular uptake similar to calcium ions [41,42]. Used in various imaging contexts, including cardiac and hepatic imaging, MEMRI is particularly effective for imaging neuronal activity. Mn^2+^ enters active neurons through voltage-gated calcium channels during functional stimulation, accumulating in brain regions and enhancing T1-weighted images. Additionally, manganese ions validate the principal eigenvector of the diffusion tensor in axonal fiber orientation studies, aligning dMRI with Mn-enhanced MRI of neural tracts.

### 4.6. Chemical Exchange Saturation Transfer (CEST)-MRI 

This high-resolution imaging technique maps specific elements in the brain by targeting exchangeable protons on molecules like OH groups. CEST-MRI selectively saturates these protons with radio-frequency pulses, transferring their magnetization to bulk water and reducing the water signal in a concentration-dependent manner. Sensitive to solute–water proton interactions at specific frequencies, CEST-MRI can detect low-concentration metabolites through their effects on the water signal without exogenous contrast agents. This method has applications in mapping pH levels and protein concentrations in the brain, exploiting amide protons (NH). In our study, CEST-MRI was applied to explore glutamate and dopamine regulation in *Unc80*-based neural disorder models, aiming to detect neurotransmission dynamics.

### 4.7. The Olfactory Habituation/Dishabituation Test 

This test evaluates mice’s ability to detect and recognize odors, leveraging their natural inclination for novelty [64]. Adult mice were exposed to filter papers infused with a first odor for 45 s over seven trials for habituation, with odor refreshment between each trial. For dishabituation, a second odor was introduced once in the same way. During the assays, the mice’s behavior was recorded with a video camera. The duration of the investigation, defined as the mouse making nasal contact with the filter paper within a 1 mm distance, was recorded. To eliminate learning biases, the test was conducted only once.

### 4.8. RNA-Sequencing 

Olfactory bulbs from CRISPR-engineered *Unc80^S/S^* and *Unc80^G/G^* mice were harvested, and RNA was extracted using TRIzol reagent (Invitrogen). cDNA libraries prepared following the TruSeq^®^ Stranded Total RNA Sample Preparation Guide (Illumina, San Diego, CA, USA, Part # 15031048) were sequenced on a NextSeq 500 (Illumina) platform, generating 75 bp paired-end reads. Quality control and primer-adaptor sequence trimming were conducted using Partek^®^ Flow^®^ Genomic Analysis Software version 10.0.23.0214 (Partek, San Diego, CA, USA), which also performed alignment to the mouse genome assembly GRCm38 (mm10). Differential expression, volcano plots, and hierarchical clustering analyses were executed using Partek’s statistical package.

### 4.9. In Silico Prediction of Unc80^WT^ and Unc80^S2367G^ Protein Structures

All protein models were generated using AlphaFold2 [35]. Structure alignment and visualization were using the PyMOL Molecular Graphics System (Ver. 2.5.7, Schrödinger, Portland, OR, USA) [36].

### 4.10. Neuro 2a Cell Culture and Transfection

Mouse Neuro 2a neuroblastoma cells were grown in MEM (Corning, Corning, NY, USA) supplemented with NEAA (Invitrogen), L-Glutamine (Invitrogen), sodium pyruvate (Invitrogen), 10% FBS (Gibco), 100 IU/L penicillin (Corning), and 10 μg/mL streptomycin (Corning) at 37 °C in a 5% CO_2_ atmosphere. Cells were transfected using TransIT X2 (Mirus, Marietta, GA, USA) following the manufacturer’s protocols. For Unc80 overexpression experiments, cells were transfected for 24 h with 2.5 μg of pcDNA3.1(−) vectors encoding GFP-tagged wild-type Unc80 or Unc80^S2732G^.

### 4.11. Indirect Immunofluorescence and Confocal Microscopy

Cells were fixed with 4% paraformaldehyde, permeabilized in 0.25% Triton X-100, and blocked in blocking buffer (5% BSA in PBS). Cells were subsequently counter-stained with DAPI in the dark for 5 min. Coverslips were mounted and analyzed by Confocal LSM780 microscopy (Zeiss, Oberkochen, Germany).

## Figures and Tables

**Figure 1 ijms-25-05985-f001:**
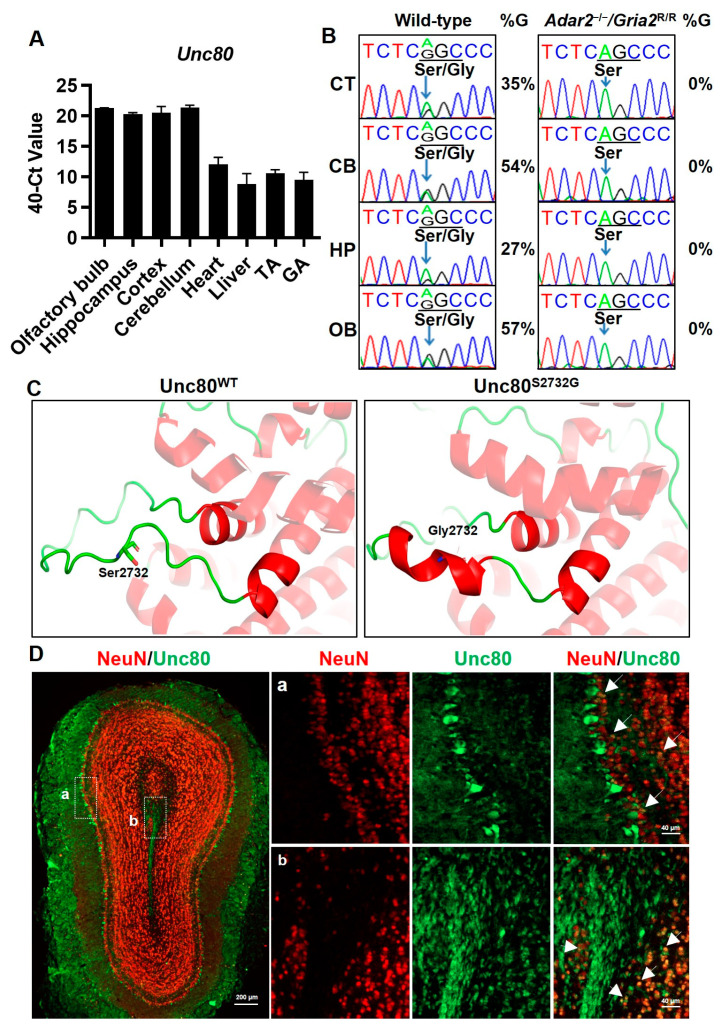
Expression patterns of ADAR2-mediated RNA editing of *Unc80* in the brain and its structural implications. Total RNA was extracted from various tissues and brain regions from wild-type (WT) and Adar2-knockout (KO) mice. The samples were then analyzed using RT-PCR (**A**) and Sanger sequencing (**B**). The sequencing chromatograms highlight the absence of guanine (“G”) signals (indicated by arrows) in the KO samples, demonstrating the reliance of *Unc80* editing on ADAR2’s enzymatic function. Differences in RNA editing levels across brain regions were also observed. The percentage represents the editing frequency, calculated by taking the peak area of G peak over the sum of A and G peaks. (**C**) Prediction of 3-dimensional protein structure models of Unc80^WT^ and Unc80^S2732G^. The magnified view of the region of interest highlights the residue change from Ser to Gly due to editing. (**D**) Immunofluorescence analysis on olfactory bulb coronal sections, specifically localizing Unc80 (green) and NeuN (red) protein. The a and b correspond to magnified views of the white dashed boxes in left panel. The arrows indicate the positions of overlapping fluorescence (green and red). Scale bars = 200 µm in the left panel and 40 µm in the magnified images. (TA: Tibialis anterior muscle, GA: Gastrocnemius muscle, CT: cortex, CB: cerebellum, HP: hippocampus, OB: olfactory bulb).

**Figure 2 ijms-25-05985-f002:**
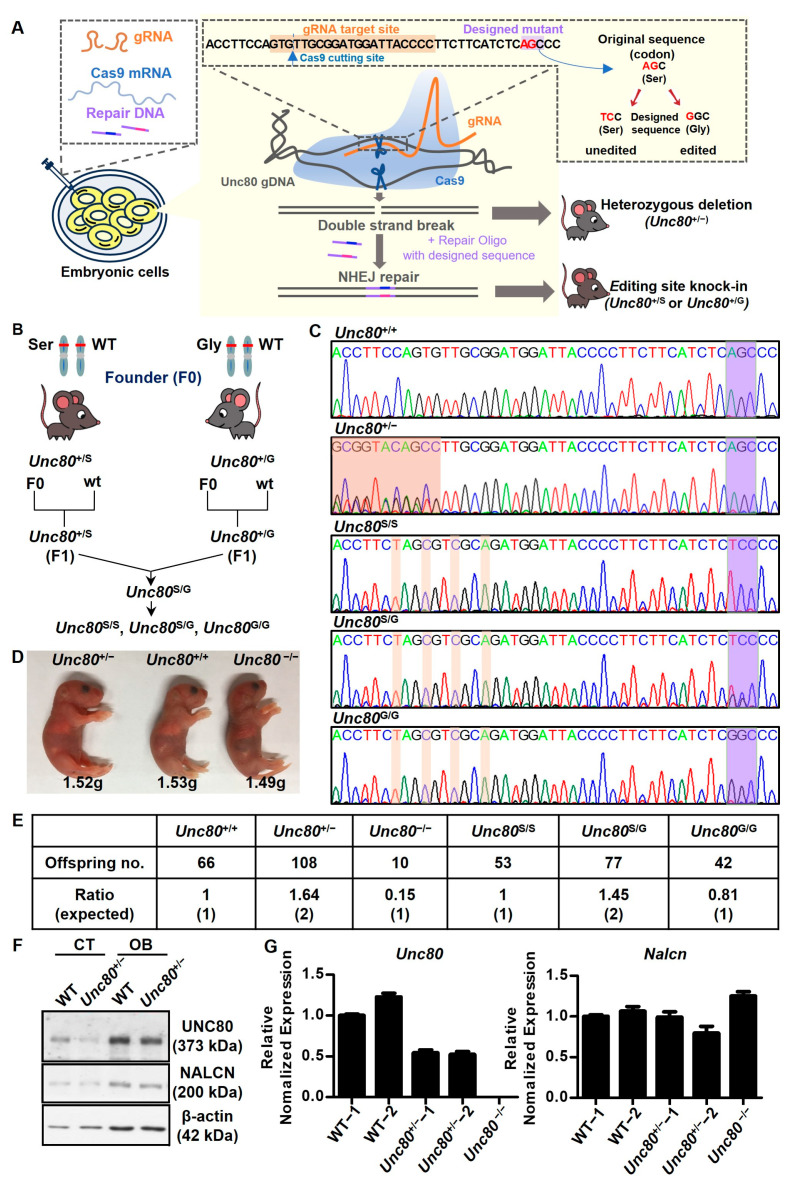
Generation of *Unc80*-deleted and site-specific RNA editing mouse models: (**A**) Experimental schematic of the CRISPR/Cas9-based genetic engineering to generate deficient and knock-in mouse models. gRNA (nucleotide with orange background) together with Cas9 created indels, thus establishing knockout mice. In parallel, the addition of synthesized homologous DNA templates corresponding to the *Unc80* sequence region with a substitution for the “pre-edited” (G-form) allele (gain of editing) or the “unedited” version (loss of editing) resulted in the knock-in models. (**B**) Breeding schemes of mice with site-specific knock-ins at the *Unc80* editing site. (**C**) Genomic DNA sequencing for genotyping demonstrates the WT sequence (top) and mutations in founder strains (*Unc80^+/−^*). Deletions and point mutations are indicated by a red background and purple boxes, respectively, in heterozygous deletion *(Unc80^+/−^*) or knock-in mice (*Unc80^S/S^*, *Unc80^G/G^*, and *Unc80^S/G^*). The green box denotes the targeted editing site. (**D**) Gross morphological comparison and body weights of newborn (P0) mice across genotypes. (**E**) Offspring genotyping from heterozygous *Unc80^+/−^* crosses reveals a lower-than-expected birth rate for homozygous deletion offspring, whereas knock-in alleles followed an approximately Mendelian inheritance pattern. (**F,G**) Unc80 protein and mRNA levels in brain tissue lysates (cortex and olfactory bulb) from CRISPR-engineered *Unc80*-deficient mice were assessed via immunoblotting (**F**) and qRT-PCR (OB, (**G**)).

**Figure 3 ijms-25-05985-f003:**
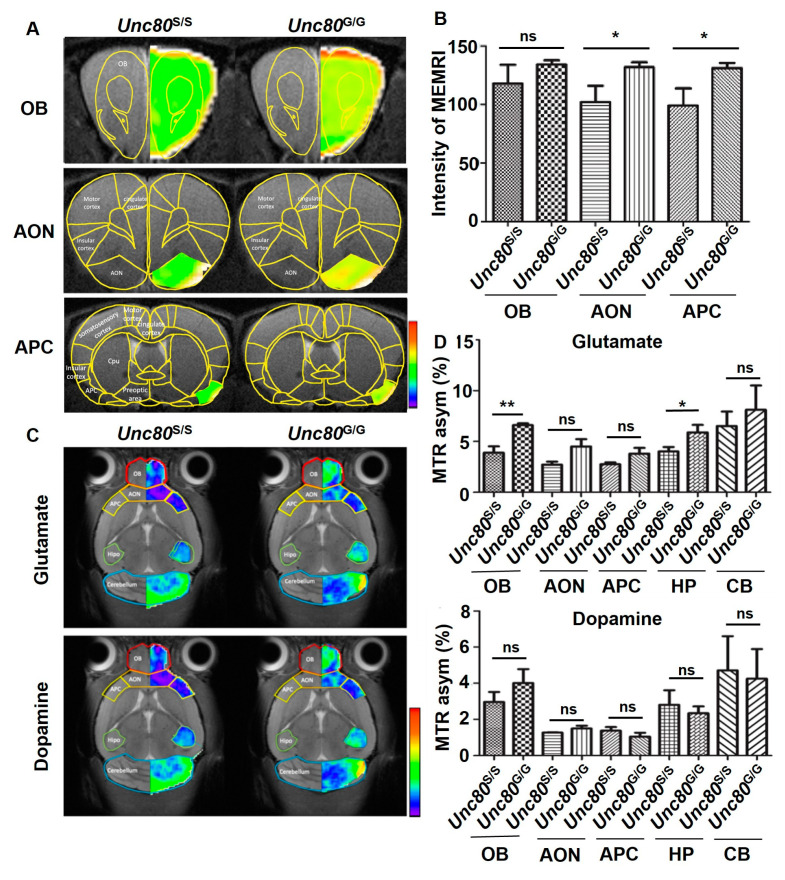
MEMRI-based neuronal activity assessment in the olfactory systems of *Unc80^S/S^* and *Unc80^G/G^* mice: (**A**) Anatomical MRI images and corresponding color mapping generated during odor stimulation. (**B**) Variation in MEMRI signal intensity in response to odor stimulation, with error bars representing the SD. (**C**) In vivo glutamate- and dopamine-sensitive CEST-MRI images showing anatomical and color mapping in various brain nuclei. (**D**) Quantitative changes in CEST-MRI signals across different brain nuclei for *Unc80^S/S^* and *Unc80^G/G^* mice, depicting both glutamate (left) and dopamine (right) contrasts. Brain regions assessed include the olfactory bulb (OB), anterior olfactory nucleus (AON), and anterior piriform cortex (APC). Error bars indicate SD. Statistical significance is denoted as follows: ns (not significant); *p* > 0.05; * *p* < 0.05; ** *p* < 0.01.

**Figure 4 ijms-25-05985-f004:**
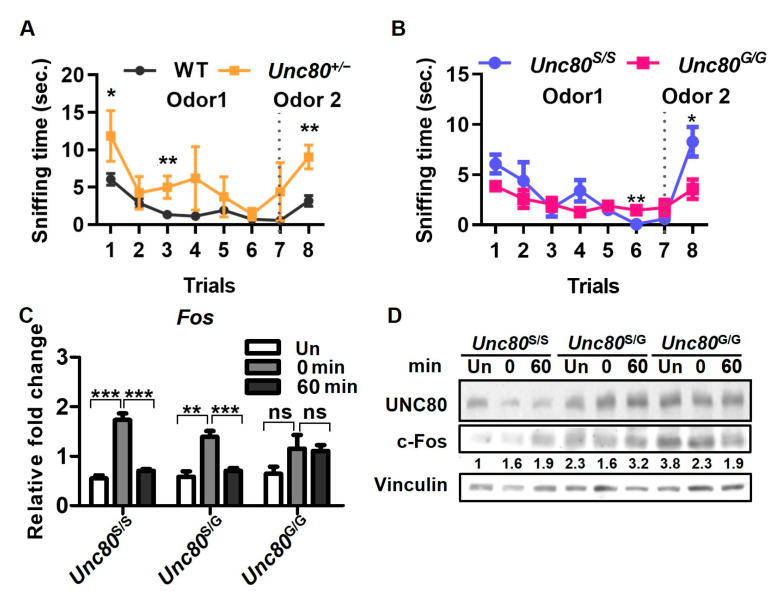
Neurophysiological connection of *Unc80* editing event to olfactory sensing and motor control: Habituation and dishabituation behaviors, in response to odors, were analyzed for *Unc80* knockout and site-specific editing variants (**A**,**B**), along with odor-induced neuronal activity assays (**C**,**D**). Mice with different genotypes (wild-type vs. *Unc80^+/−^* in (**A**), *Unc80^S/S^* vs. *Unc80^G/G^* in (**B**)) were exposed to odors, and their explorative times near the odor source were recorded and presented as mean ± SD. The cohorts consisted of: WT (n = 13), *Unc80^+/−^* (n = 5), *Unc80^S/S^* (n = 9), *Unc80^G/G^* (n = 8). For the neuronal activation assay, knock-in mice with site-specific edits were either not exposed or exposed to banana oil for up to 60 min and subsequently sacrificed for olfactory bulb isolation, from which total RNA and proteins were prepared for qRT-PCR (**C**) and Western blot (**D**) analyses, respectively. Changes in the expression of c-Fos were monitored as a readout for neuronal activation. The bar graph represents the relative mRNA expression levels of Fos. Statistical significance is indicated as follows: ns (not significant); *p* > 0.05; * *p* < 0.05; ** *p* < 0.01; *** *p* < 0.001.

**Figure 5 ijms-25-05985-f005:**
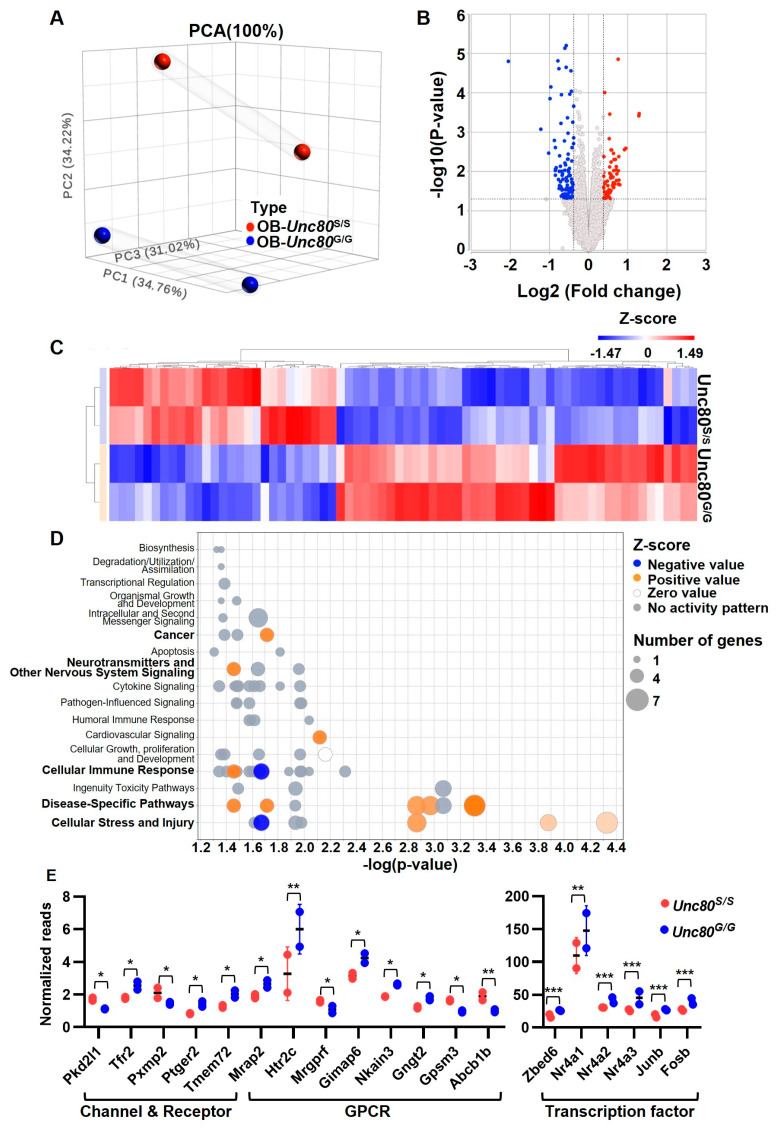
Transcriptomic analysis of the olfactory bulb in *Unc80* editing variant mice. Transcriptome-wide RNA-seq was conducted to identify changes in the olfactory bulb of *Unc80^S/S^* and *Unc80^G/G^* mice. (**A**–**C**) The overall transcriptome distribution is represented in a principal component analysis (PCA) plot (**A**) and a volcano plot (**B**), delineating genotype-specific gene expression profiles. A heatmap (**C**) displays genes with significant differential expression (|fold-change| > 1.5, *p* < 0.05; n = 6 per genotype, with samples pooled from three mice each) between the two strains. (**D**) Bubble chart of the enriched canonical pathway from IPA analysis. The *x*-axis represents the significance (*p*-value), and the *y*-axis shows the top 17 significantly enriched pathways based on differentially expressed genes (DEGs). The size of the circles corresponds to the number of genes associated with each pathway. The colors of the bubbles denote the Z-score, indicating whether the pathway is activated (orange, positive Z-score) or inhibited (blue, negative Z-score). (**E**) Normalized read count plots highlight expression variations in gene sets linked to neuronal signaling between *Unc80^S/S^* and *Unc80^G/G^* mice. Statistical significance in this figure is indicated as follows: * *p* < 0.05; ** *p* < 0.01; *** *p* < 0.001.

## Data Availability

The data supporting the findings of this study are publicly available in the GEO database (accession number: GSE269152).

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
