# Peer review of "Imbalance in Unc80 RNA Editing Disrupts Dynamic Neuronal Activity and Olfactory Perception"

_ijms, 2024, doi:10.3390/ijms25115985_

Round 1

Reviewer 1 Report

Comments and Suggestions for Authors

The current manuscript represents an interesting genetics approach to investigate the phenomenon of RNA Editing. Although the models may not be fully physiological, they provide some insights into the consequences of RNA editing at a single gene. The following are my suggestions to improve the manuscript:

Major:

1. The abstract needs to be simplified: All genotypes need to discussed.

2. Line 148-149: The methods for measuring neurotransmitters are not clear and how they can be validated is also not explained.

3. What are the units for neurotransmitter measurements and what are the distributions of these values? Until these are clarified, I don't think talk about statistics is appropriate.

4. RNA-seq data presentation is of relatively low quality: labels on the heatmap are not needed or it has to be modified. Enrichments of  categories of genes need to be shown.

5. Lines 284-286: Over-interpretation of the data. What patterns are you talking about?

Minor:

1. Line 52: What is Gri2a gene? It is mentioned but without connection to the previous text.

2. Lines 55-56: change usage of "involved".

3. Line 103: Please define "editome" and explain the process.

4. Lines 328-329: You can't call a modest RNA-seq experiment "comprehensive".

Reviewer 2 Report

Comments and Suggestions for Authors

Chen et al.,

The manuscript by Chen et al., investigates the functional relevance of an evolutionary conserved editing site within the UNC80 gene. Using state of the are genetic tools to generate edited and unedited animals they demonstrate that the editing state of UNC80 affects olfactory behavior and neuronal excitability.

The manuscript is well written, and most of the results are well presented. Additional experiments are required to support the mechanism suggested by the athors.

Major Comments:

1.    In Fig.1A, it is hard to understand what the relative fold change is. It is not clear what is the editing level of this particular site, and specifically what is the editing level at the OB, AON and APC.

2.    Given the recoding of Ser2732 to Gly, it is important to show the location of the recoding site within the context of the domain structure or Alpha-fold based model that can help explaining the way by which editing is related to the function of the protein.

3.    The explanation regarding the approach used to generate the knock in edited and unedited animals is not clear, please provide schematics. In addition, the logics behind generating a K.O line and comparing it to the knock in lines is not clear, and to my opinion distract from the main question that focuses on RNA editing.

4.    It is possible that the genetic manipulation of the coding seq affects proper localization of the protein, the bulk expression presented is not enough.

5.    To prove the role of editing within OB neuron, one can rescue the phenotype by expressing an edited version.

6.    The logics behind performing a transcriptomic analysis of the different lines in not straight forward, and the contribution of this data to the mechanistic aspect of the story is not clear.

Minor comments:

7.    Sentence written in lines 67-68 is not clear.

8.    There are many places within the introduction and discussion that lack references. Examples are in lines: 59, 64, 103. Please go over the entire manuscript and make sure that you reference the facts properly.

9.    Line 72, please reference the original study.

Round 2

Reviewer 2 Report

Comments and Suggestions for Authors

The revised version is considerably improved